# Proteomics Characterization of Clear Cell Renal Cell Carcinoma

**DOI:** 10.3390/jcm12010384

**Published:** 2023-01-03

**Authors:** Jesús Miranda-Poma, Lucía Trilla-Fuertes, Rocío López-Vacas, Elena López-Camacho, Eugenia García-Fernández, Ana Pertejo, María I. Lumbreras-Herrera, Andrea Zapater-Moros, Mariana Díaz-Almirón, Antje Dittmann, Juan Ángel Fresno Vara, Enrique Espinosa, Pilar González-Peramato, Álvaro Pinto-Marín, Angelo Gámez-Pozo

**Affiliations:** 1Medical Oncology Service, Hospital Universitario Quironsalud Madrid, 28223 Madrid, Spain; 2Molecular Oncology Laboratory, Hospital Universitario La Paz—IdiPAZ, 28046 Madrid, Spain; 3Biomedica Molecular Medicine SL, 28049 Madrid, Spain; 4Department of Pathology, University Hospital La Paz, 28046 Madrid, Spain; 5Medical Oncology Service, Hospital Universitario La Paz, 28046 Madrid, Spain; 6Biostatistics Unit, Hospital Universitario La Paz—IdiPAZ, 28046 Madrid, Spain; 7Functional Genomics Center Zurich, 8057 Zurich, Switzerland; 8Biomedical Research Networking Center on Oncology-CIBERONC, ISCIII, 28029 Madrid, Spain; 9Cátedra UAM-Amgen, Universidad Autónoma de Madrid, 28049 Madrid, Spain

**Keywords:** proteomics, molecular subtypes, clear cell renal cell carcinoma

## Abstract

Purpose: To explore the tumor proteome of patients diagnosed with localized clear cell renal cancer (ccRCC) and treated with surgery. Material and methods: A total of 165 FFPE tumor samples from patients diagnosed with ccRCC were analyzed using DIA-proteomics. Proteomics ccRCC subtypes were defined using a consensus cluster algorithm (CCA) and characterized by a functional approach using probabilistic graphical models and survival analyses. Results: We identified and quantified 3091 proteins, including 2026 high-confidence proteins. Two proteomics subtypes of ccRCC (CC1 and CC2) were identified by CC using the high-confidence proteins only. Characterization of molecular differences between CC1 and CC2 was performed in two steps. First, we defined 514 proteins showing differential expression between the two subtypes using a significance analysis of microarrays analysis. Proteins overexpressed in CC1 were mainly related to translation and ribosome, while proteins overexpressed in CC2 were mainly related to focal adhesion and membrane. Second, a functional analysis using probabilistic graphical models was performed. CC1 subtype is characterized by an increased expression of proteins related to glycolysis, mitochondria, translation, adhesion proteins related to cytoskeleton and actin, nucleosome, and spliceosome, while CC2 subtype showed higher expression of proteins involved in focal adhesion, extracellular matrix, and collagen organization. Conclusions: ccRCC tumors can be classified in two different proteomics subtypes. CC1 and CC2 present specific proteomics profiles, reflecting alterations of different molecular pathways in each subtype. The knowledge generated in this type of studies could help in the development of new drugs targeting subtype-specific deregulated pathways.

## 1. Introduction

Renal cell carcinoma (RCC) is the sixth most common cancer in men and the eighth most common cancer in women. It is estimated that in 2022 there will be 79,000 new cases and that it will cause 13,920 deaths in the United States [1].

Two-thirds of patients have localized disease and an additional 16% have locoregional disease at diagnosis. A significant proportion of all these patients (up to 40% in stage III) will relapse [2,3].

Nowadays, there are many drugs approved for the treatment of clear cell renal cancer (ccRCC); most of them belong to the group of VEGFR inhibitors (Sunitinib, Pazopanib, Axitinib, Lenvatinib) or to the group of immune checkpoint inhibitors (Pembrolizumab, Nivolumab, Atezolizumab, Ipilimumab). Despite their demonstrated effectiveness in advanced disease [4,5,6,7,8], modest results have been showed in the adjuvant setting [9,10]. In any case, when these treatments fail, few therapeutic options remain. Therefore, research into potential new therapeutic targets is a need today.

The molecular characterization of ccRCC is an essential tool to acquire a deeper and more precise knowledge of the different alterations involved in the carcinogenesis process, of its capacity to develop metastasis and also to identify possible therapeutic targets.

Genomic profiling techniques, such as next-generation sequencing (NGS), are widely used to identify DNA sequences altered by somatic mutations that are associated with carcinogenesis. However, DNA sequencing and the finding of existing mutations provides little information about the functional consequences of these mutations. In the same way, transcriptomics analyses are used to assess gene expression by measuring coding mRNA transcripts; however, although they reflect the functional modules that regulate cellular processes, they provide little information about the physiological reality of the cell in comparison with proteomics. Additionally, there is increasing evidence that mRNA transcript expression and protein expression do not correlate robustly in normal tissues or tumors [11,12].

The use of proteomics profiling can contribute to a better understanding of the process of carcinogenesis and also may be helpful in daily clinical practice, such as, for example, in diagnosis, through the determination of specific ccRCC proteins [13], as serum biomarkers for follow-up, such as those proposed in the study by White et al. [14], as a prognostic factor such as the NDRG1 protein [15], to identify patients at greater risk of relapse [16], and also to determine the most effective treatment in each case [11,17].

Different proteomics profiles have also been found for patients with localized and metastatic ccRCC. Masui et al. [18], found that three proteins (LGALS1, PFN1 and YWHAZ) showed higher expression in primary ccRCC compared to normal adjacent tissue (NAT), and higher expression in metastatic lesions than in primary tumors. In addition, PFN1 was shown to be associated with poor prognosis.

In the field of ccRCC, the most relevant studies that analyzed proteomics profiles are: in western populations, the study performed by the clinical proteomics tumor analysis consortium (CPTAC) [17], and in eastern populations, the study performed by Qu et al. [16]. Both studies identified ccRCC subtypes with differences in the predominant altered pathways and in some cases with different survival.

In the CPTAC study [17], 103 samples from patients diagnosed with ccRCC were analyzed and compared with 84 NAT samples. At the proteomics level, when comparing ccRCC samples with NAT samples, 820 proteins showed significant differential expression, with 565 proteins downregulated and 255 upregulated. Enrichment analysis revealed that the immune response, epithelial mesenchymal transition (EMT) and multiple signaling pathways (hypoxia, glycolysis and mTOR) were upregulated in tumors, and that the tricarboxylic acid cycle (TCA), also known as the Krebs cycle, fatty acid metabolism and OXPHOS were downregulated.

Furthermore, by combining the analysis of transcriptomic profiles (gene signatures) of immune, stromal and microenvironmental cells with proteomics features, the group defined four main subtypes of ccRCC. These four subtypes are: CD8 (+) inflammatory, CD8 (−) inflammatory, VEGF/immune-deserted, and immune-deserted metabolic tumors. These subtypes not only predicted the response to different treatments such as immune checkpoints and anti-VEGF therapies, but also predicted patients’ survival [11,17].

Inflamed CD8 (+) tumors were characterized at the proteomics level by upregulation of CD38 and pathways involved in antigen processing/presentation (APM). Inflammatory CD8 (−) tumors displayed as a unique feature for this subtype an elevated PDGFRA, abundance of extracellular matrix (ECM)-associated proteins, and epithelial mesenchymal transition (EMT). Immune-desert metabolic tumors had increased mTOR signaling and a unique metabolic profile including elevated expression of mitochondrial proteins, OXPHOS and glycolysis [17].

In the study performed by Qu et al. [14], an exhaustive proteogenomic analysis of 232 pairs of adjacent ccRCC and NAT tumors was performed. They used consensus clustering to identify ccRCC proteomics subtypes, and classified ccRCC into three subtypes, defined by the authors as GP1, GP2 and GP3. These proteomics subtypes exhibited differences in overall survival (OS) and progression free survival (PFS). Among the three subtypes, GP1 was shown to have the highest mortality risk.

The GP1 subtype was characterized by a high degree of immune infiltration, including innate immune system, complement and coagulation cascades, antigen processing and cross-presentation, interferon signaling and T-cell receptor (TCR) signaling, all of these alterations resulting in it being the most immunosuppressed subtype. The GP2 subtype showed increased metabolism-related pathways including the TCA cycle and respiratory chain, amino acid metabolism, mitochondrial translation, lipid metabolism and glycolysis/gluconeogenesis, and the GP3 subtype had the highest stromal scores, corresponding to upregulation of extracellular matrix (ECM)-related pathways, including ECM organization, collagen formation, elastic fiber formation and focal adhesion.

The aims of the study was to explore the tumor proteome of patients diagnosed with localized ccRCC and treated with surgery and the identification of altered pathways that could be the target of new treatments.

## 2. Materials and Methods

### 2.1. Patient Clinical Characteristics

Formalin-fixed paraffin-embedded (FFPE) samples from patients diagnosed with ccRCC were recruited from the Hospital La Paz Biobank, Basque Biobank and Biobank of Servicio de Salud Andaluz. The cohort included a total of 165 patients with diagnosis of clear cell renal cell carcinoma, stage pT1b-pT3, Nx, M0, with any Furhman grade (Table 1). Written informed consent was obtained for each participant and the Hospital La Paz Ethics Committee approved this study (PI3310).

### 2.2. Sample Processing and Protein Isolation

Protein isolation was performed as previously described [19]. Briefly, FFPE sections were deparaffinized in xylene and washed twice in absolute ethanol. Protein isolates were prepared in 2% of SDS. Protein quantity was measured using MicroBCA Protein Assay Kit (Pierce-Thermo Scientific, Langenselbold, Germany). Finally, 10 µg of each protein extract were digested with trypsin (1:50) and SDS was eliminated from the lysates using Detergent Removal Spin Columns (Pierce, Langenselbold, Germany). Before mass-spectrometry experiments, samples were desalted using ZipTips (Millipore, Burlington, MA, USA), dried, and resolubilized in 15 µL of a 0.1% formic acid and 3% acetonitrile solution.

### 2.3. DIA Proteomics Experiments

Peptides were cleaned up using C18 stage-tips, re-solubilized in MS sample buffer and spiked with indexed retention time (iRT) peptides. Ten pools of all samples were used for data-dependent acquisition (DDA) runs. Mass spectrometry analysis was performed on an Orbitrap Fusion (Thermo Scientific, Langenselbold, Germany) equipped with a Digital PicoView source (New Objective) and coupled to an M-Class UPLC (Waters), operated in trapping mode. Peptides were loaded onto a commercial MZ Symmetry C18 Trap Column (5 µm, 180 µm × 20 mm, Waters) followed by nanoEase MZ C18 HSS T3 Column (1.8 µm, 75 µm × 250 mm, Waters). The peptides were eluted at a flow rate of 300 nl/min, with a gradient from 5% to 22% for 109 min. The DDA and DIA runs were acquired in Orbitrap-Orbitrap mode with isolation windows of 1.4 m/z (DDA, cycle time 3 s) and 20 m/z covering a range from 400 to 1100 m/z (DIA). Spectronaut 14 was used in conjunction with the Pulsar search engine to generate a hybrid spectral by applying the default parameter settings to DDA and DIA runs. Spectra were searched against a canonical SwissProt database for human and common protein contaminants (NCBI taxonomy ID9606, release date 9 July 2019). Protein quantification was performed in Spectronaut using the default settings. The quantitative data were extracted using the BGS Factory Report (default) and used for follow-up analyses. To perform statistical modeling, fragment intensities were aggregated into precursor and peptide intensities. A quality criterion of at least 75% of valid values was applied and missing values were imputed to a normal distribution using Perseus software [20]. Proteomics data is available at PRIDE repository (http://www.ebi.ac.uk/pride).

### 2.4. Statistical Analyses

Proteomics ccRCC subtypes were defined using the processing proteomics data employing a consensus cluster algorithm (CC) [21]. The CC was performed in R environment using the *ConsensusClusterPlus* package [22]. The optimum number of groups was determined by the delta plot and CC was calculated using Eucledian distance, average as linkage method and k-means as the cluster algorithm.

Comparisons between groups were performed using a non-parametric Mann–Whitney test and survival analysis was performed using a Kaplan–Meier and log-rank test. All *p*-values are two-sided and considered significant under 0.05. Statistical analyses were performed in GraphPad Prism version 6. Significance analysis of microarrays (SAM) was performed using TM4 Mev software [23]. SAM allows the identification of differential proteins, assigning a score to each protein based on the change in protein expression relative to the standard deviation of repeated measurements [24]. SAM employs permutations of the repeated measurements to estimate the percentage of proteins identified by chance, the false discovery rate (FDR). In this case, an FDR below 5% was fixed.

### 2.5. Systems Biology Analyses

A network using proteomics data without other a priori information was built. For that, probabilistic graphical models (PGMs) compatible with high-dimensional data were used, as previously described [19]. Briefly, the network was built in two sequential steps: first, the spanning tree with the maximum likelihood was determined, and then, the simplest graph with edges that reduce Bayesian information criterion (BIC) and preserve decomposability was built [25]. PGMs were calculated using R environment and *grapHD* package [26].

Functional structure of the PGM was established using DAVID webtool, with “Homo sapiens” as background and GOTERM-FAT, KEGG and Biocarta as categories [27]. To make comparisons between groups of tumors, functional node activities were calculated as the mean expression of those proteins related to the main function of each branch.

## 3. Results

### 3.1. DIA Proteomics Experiments

A total of 3091 proteins were identified in 165 ccRCC paraffin samples by DIA proteomics experiments. After applying quality criteria (at least 75% of valid values across the sample series), 2026 proteins were used for the analyses.

### 3.2. Definition of ccRCC Proteomics Subtypes

With the aim of classifying samples, CC was applied to the proteomics data. Two groups of ccRCC patients were defined according to their proteomics profile. CC1 included 55 patients (33%) and CC2, 110 patients (67%). SAM analysis identified 514 proteins differentially expressed between these two groups (Figure 1). Proteins overexpressed in CC2 were mainly related to focal adhesion and membrane while proteins overexpressed in CC1 were mainly related to translation and ribosome.

There was no significant difference according to disease-free survival (DFS) or overall survival (OS) between the two ccRCC proteomics subtypes (Figure 2 and Figure 3).

### 3.3. Systems Biology Analysis of ccRCC Proteomics Data

A system biology analysis based on PGMs was performed using proteomics data. The resulting network was divided into eight functional nodes with an overrepresented function: complement activation, focal adhesion, splicing, translation, nucleoplasm, mitochondria, and glycolysis (Figure 4A). Functional node activities were used to determine biological processes with a different activity between the two proteomics subtypes. There were significant differences in all the nodes with the exception of the complement activation node. Adhesion node activity was higher in CC2 and spliceosome, adhesion 2, translation, nucleosome, mitochondria, and glycolysis had a higher activity in CC1.

The complement node was formed by immunoglobulins and complement proteins. The translation node was composed of ribosomal proteins. The adhesion node had proteins related to focal adhesion, extracellular matrix, and collagen organization while adhesion 2 node had proteins involved in cytoskeleton, actin and cadherin binding. The mitochondria functional node contained proteins involved in mitochondrial metabolism such as OGDH, PDHA1, NDUFS1, or SDHA. The glycolysis functional node was composed of proteins involved in glucose metabolism: ALDOA, PGK1, PFKP, ENO1, GPI, LDHA, PKM, etc. The nucleosome functional node contained some proteins involved in chromatin processes such as SMARCA4 or PARP1.

## 4. Discussion

In this study, a proteomics characterization of 165 localized ccRCC samples coupled with a system biology approach allow us the identification and characterization of two ccRCC proteomics subtypes (CC1 and CC2). A differential expression of 514 proteins was found between the two groups.

In functional analysis we found that the CC1 subtype was characterized by an increase in the expression of proteins related to glycolysis, mitochondria, translation, and adhesion, specifically proteins related to the cytoskeleton and actin.

It is known that in normal cells under normoxic conditions, glucose is a major source of pyruvate, which feeds the tricarboxylic acid (TCA) cycle for energy production, but under hypoxic conditions, normal cells shift their energy production from the TCA cycle to lactate fermentation. In contrast, in ccRCC cells, energy is predominantly produced by lactic acid fermentation, regardless of oxygen level. This shift or reprogramming in metabolism is known as the Warburg effect or aerobic glycolysis [28]

In order to understand how this metabolic reprogramming occurs, we start from the knowledge that the loss of function in the von hippel lindau (VHL) gene is a common phenomenon in ccRCC. In VHL-deficient ccRCC, hypoxia inducible factor 1α (HIF-1α) increases, which in turn increases the expression of glucose transporter 1 (GLUT-1), which promotes cellular glucose uptake. HIF-1α also upregulates the expression of genes encoding enzymes involved in glycolysis, such as hexokinase 1 and 2, glyceraldehyde 3-phosphate dehydrogenase (GAPDH) and pyruvate kinase (PKM). In addition, HIF-1α upregulates the expression of Lactate Dehydrogenase A (LDH-A) and thus promotes the conversion of pyruvate to lactate and shifts cellular metabolism out of the TCA cycle by regulating pyruvate dehydrogenase [29]. In our work on the glycolysis node, we found overexpression of several proteins including GAPDH, PKM and LDHA-A.

Interestingly, increased GLUT-1 expression in ccRCC tumors, is correlated with a decrease in the number of infiltrating CD8+ T cells [29,30], suggesting an additional mechanism by which ccRCC might suppress the immune system.

HIF-1α also regulates the expression of several microRNAs, including miR-210 [31], which is overexpressed in ccRCC and has been shown to downregulate mitochondrial respiration.

In line with the importance of this pathway, we have to mention that Belzutifan [32], an HIF-2α antagonist, has been recently approved by the US Food and Drug Administration (FDA) for the treatment of ccRCC. The CC2 subtype was characterized by a higher expression of proteins related to the focal adhesion process (the process by which the actin microfilament cytoskeleton is anchored intracellularly to extracellular matrix proteins).

This adhesion node had proteins related to focal adhesion, extracellular matrix, and collagen organization. Cell adhesion plays a key role in the development of metastasis, altered expression of genes involved in adhesion and remodeling of the extracellular matrix (ECM) causes changes in the contacts between neighboring cells and between cells and the ECM, contributing to the metastatic process [33].

Wang et al. [34], compared RCC tumor tissue with NAT and found a higher expression of proteins related to cell adhesion in the tumor tissue; and a group of these proteins that were associated with a worse prognosis and higher risk of progression such as CD44, CD86, Fibronectin 1 (FN1), Integrin Subunit Alpha M (ITGAM), and Integrin Subunit Beta 2 (ITGB2). ITGAM and ITGB2 are subunits of integrin, and these cell adhesion-related molecules are closely related to cancer cell invasion and metastasis. Further studies are needed to determine their exact role in RCC. In our study we found overexpression of ITGA6, which belongs to the same family as the previous mentioned proteins.

Other proteins that also showed increased expression in tumor tissue are Protein Tyrosine Phosphatase Receptor Type C (PTPRC) and Toll-Like Receptor 2 (TLR2). However, their exact mechanisms are not yet fully understood in RCC [33]. In our work we also found an overexpression of PTPRC.

Also, the alteration of this cell adhesion mechanism is involved in the epithelial mesenchymal transition (EMT) and seems to also contribute to the resistance of tumor cells to chemotherapy and radiotherapy [35].

The two major studies on proteomics are the one performed by CPTAC [17] and the work performed by Qu’s group [16]. In both studies different ccRCC subtypes were identified according to their proteomic profile. In addition, Qu’s group correlated their classification with the classification established by the CPTAC, finding that the GP1 subtype was mainly CD8 (+) inflammatory tumors, the GP2 subtype was mainly immune-deserted metabolic subtype tumors, and the GP3 subtype was mainly CD8 (−) inflammatory tumors.

In contrast to the previously mentioned studies, which included patients with localized and advanced disease, our work was focused on the group of patients with localized disease. When we compared our results with those of the CPTAC [17], we found that our CC1 subtype, which is mainly characterized by alterations in metabolic pathways (glycolysis, mitochondrial alteration), is similar to the subtype called metabolic-immune-desert, that presents alterations in the same pathways. While our CC2 subtype is similar to the CD8 (−) inflammatory tumor subtype, as both at the proteomics level present alterations in pathways that regulate processes linked to the extracellular matrix (ECM), to the focal adhesion process and to EMT.

Whereas if we compare our proteomics subtypes with those described by Qu et al. [16], it is possible to intuit that our CC1 subtype is equivalent to the GP2 subtype, because both groups present the alteration of metabolic processes such as glycolysis and cellular respiration as their most important characteristic. Also, our subtype CC2 could be equivalent to GP3, as both groups present the alteration of processes related to the extracellular matrix, including the focal adhesion process, as a main characteristic. We did not identify a subtype equivalent to GP1 because it includes a high proportion of stage IV tumors, which were not analyzed in this work.

In our work, when we compared the progression-free survival (PFS) obtained by each subtype, we found that there is no statistically significant difference between them. The same situation occurred when we compared OS between the two subtypes, with no differences. These results are similar to the results reported by Qu et al. [16], who did not find significant differences in DFS or OS between the GP2 and GP3 subtypes.

Although our results are largely in agreement with the results obtained by CPTAC [17] and Qu’s group [16], in our work we did not identify clusters with a predominance of alterations in the immunity pathway. This difference may be explained by the fact that our work is only a proteomics analysis whereas in the previously mentioned studies both proteomics and transcriptomics data were combined for the identification of their different subtypes. Another reason for this difference between the previously mentioned papers [16,17] and ours is that they compared tumor tissue with NAT; while we compared only tumor tissue samples.

Nevertheless, we consider that the proteomics-only study also provides valid data for the identification of the predominant altered pathways in each subtype, as well as the usefulness of these proteomics profiles in the search for possible therapeutic targets.

For example, the glycolysis pathway alteration is a characteristic of our CC1 subtype (metabolic); today we can find several drugs that are accumulating growing evidence that they could have a role in controlling this pathway. Some examples are:Dichloroacetate (DCA), already used to treat acute and chronic lactic acidosis, inborn errors of metabolism and diabetes, these small molecules selectively target cancer cells and switch their metabolism from glycolysis to oxidative phosphorylation; but their clinical administration in cancer therapy is still limited to early phase clinical trials [36,37].2-deoxy-D-glucose (2-DG) is one of the most effective anti-glycolytic agents. It is phosphorylated by hexokinase (HK), which is the first rate-limiting enzyme of glycolysis and subsequently inhibits the pentose-phosphate pathway (NAPDH) and ATP generation [38]. This molecule is one of the most studied nowadays, because if we search in the clinical trials database we can find 37 open trials, all of them in early phases.

With these examples we have tried to demonstrate that knowing the different altered metabolic pathways (glycolysis, cellular respiration, focal adhesion) in ccRCC can guide us in the search for new treatments.

Our group believes that the proteomics study could provide interesting information for a better understanding of the carcinogenesis process, for better decision making in daily clinical practice and also for the development of new treatments.

## 5. Conclusions

In the present work, using proteomics analysis we found two distinct subtypes of ccRCC, differentiated by the alteration of distinct molecular pathways. CC1 is characterized by the alteration of metabolic pathways, translation, and adhesion related to cytoskeleton, spliceosome, and nucleosome, whereas CC2 was characterized by the alteration of focal adhesion, extracellular matrix and collagen organization. These results are consistent with those found in other previously cited studies. The importance of this type of study is to know more about the altered molecular pathways because this provides us with more knowledge about them and may help us in the development and research of new drugs that act on these pathways.

## Figures and Tables

**Figure 1 jcm-12-00384-f001:**
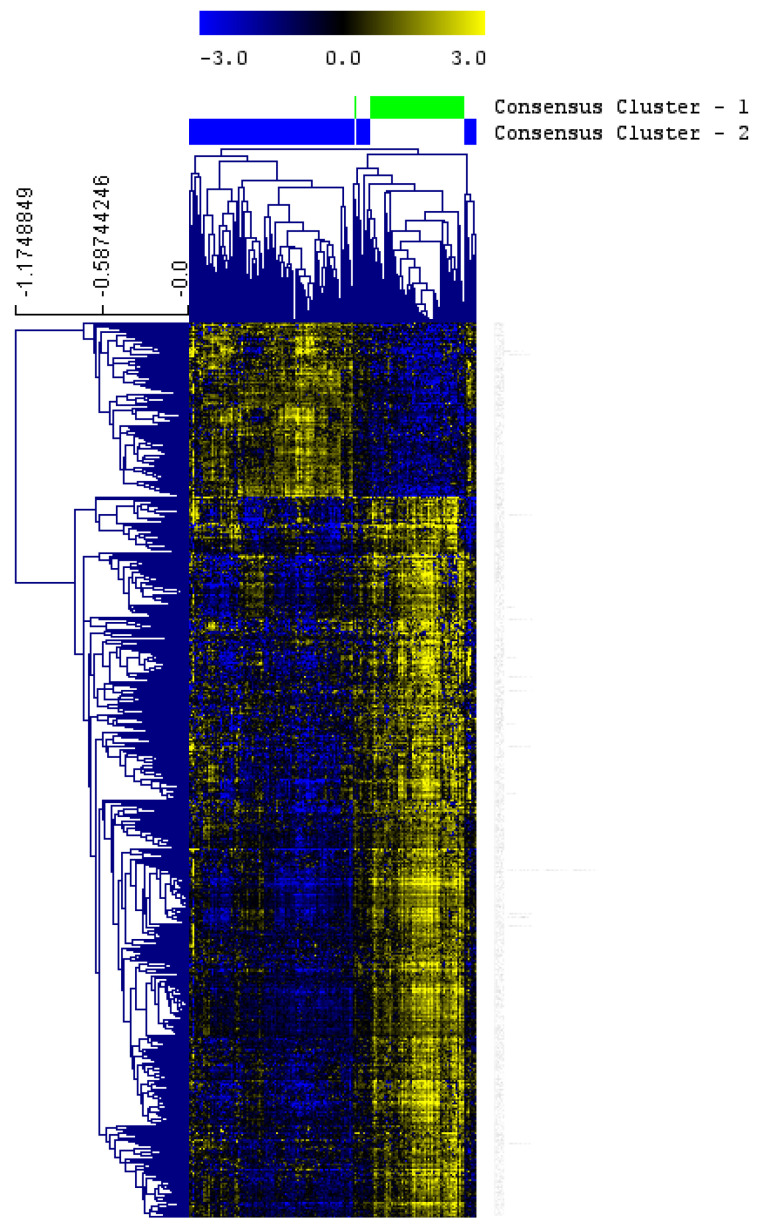
Proteins differentially expressed between the two ccRCC proteomics subtypes identified by a significance analysis of microarrays. HCL dendrogram was built using average linkage. Blue = underexpressed. Yellow = overexpressed.

**Figure 2 jcm-12-00384-f002:**
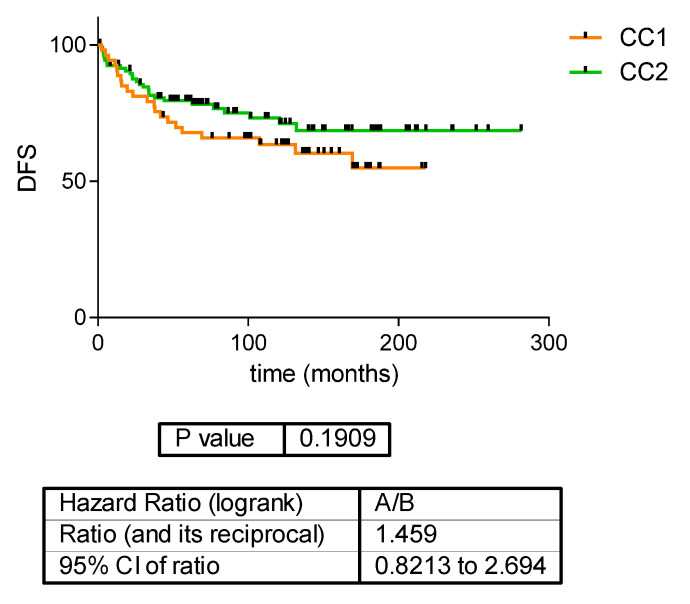
Disease-free survival in the two ccRCC proteomics subtypes.

**Figure 3 jcm-12-00384-f003:**
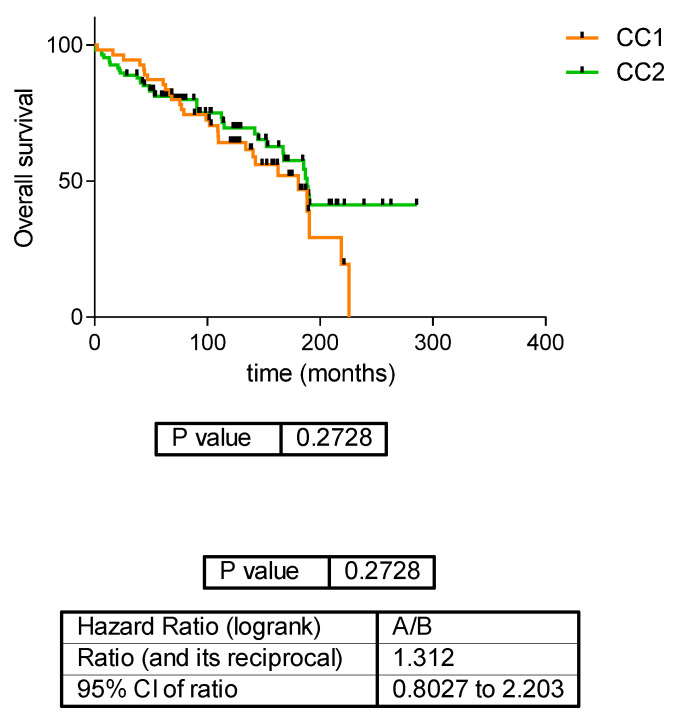
Overall survival in the two ccRCC proteomics subtypes.

**Figure 4 jcm-12-00384-f004:**
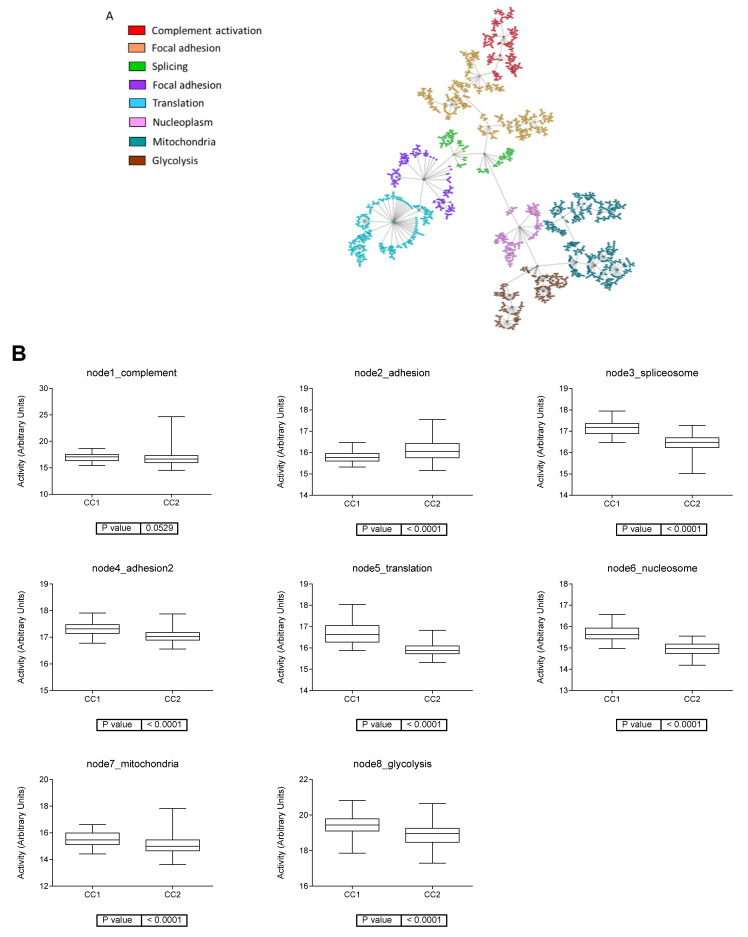
Systems biology analysis of ccRCC proteomics data. (**A**). Network obtained in the probabilistic graphical model analysis from the proteomics data. (**B**). Boxplots of functional node activities comparing the two proteomics ccRCC groups.

**Table 1 jcm-12-00384-t001:** Clinical Characteristics of the Patients.

Furhman grade	1	20	12%
	2	79	48%
	3	53	32%
	4	12	7%
	Unknown	1	1%
T stage	T1b	75	45%
	T2	18	11%
	T3	72	44%

## Data Availability

Proteomics data analysed in this work will be included in a series of papers assessing other aspects of RCC management, such as risk of recurrence prediction. We plan to make the data available at PRIDE (http://www.ebi.ac.uk/pride) after all these papers get published.

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
