# Peer review of "Proteomics Characterization of Clear Cell Renal Cell Carcinoma"

_jcm, 2023, doi:10.3390/jcm12010384_

Round 1

Reviewer 1 Report

The authors found that ccRCC tumors can be classified in two different proteomics subtypes, and CCA1 and CCA2 represent specific proteomics profiles, reflecting alterations of different molecular pathways in each subtype. These findings add new knowledge of subtype-specific deregulated pathways in ccRCC. However, there is no significant difference according to DFS or OS between the two ccRCC subtypes, and they did not show whether the two subtypes were related to the grades and stages of ccRCC, suggesting the classification may be not relevant ot the clinical treatment of this disease. Moreover, the findings of this study contain some points already revealed by the previous reports. Overall, this sudy is not of importance and lack of novelty.

Reviewer 2 Report

In this paper, proteomic analysis was performed on FFPE samples from ccRCC patients to divide the patients into two clusters according to their protein profiles and to determine what characteristics are present in each cluster. Although there were no significant differences in survival curves between the two clusters, it is hoped that such an analysis will improve our understanding of the molecular basis for the development and progression of ccRCC. I have listed below the points that I believe need improvement. I would like to ask the authors to address these points to improve the paper.

Major points:

・In CCA, there is no description on the type of algorithm (e.g. hierarchical clustering, ward.D2) or the definition of distance (e.g. Euclidean). The process by which the number of clusters was determined to be optimal at 2 should be described. For example, consensus matrix, consensus cumulative distribution function plot, delta area plot or item-consensus plot. Similarly, in Fig. 1, there is no information on the type of hierarchical clustering algorithm shown in the dendrogram, or on the definition of distances. The information on the conditions for SAM analysis is lacking either.

・Their results are compared with those of Qu et al, but the explanation of GP1-3 is not sufficient. I cannot understand the meaning of “GP” without checking the original paper cited, so please add a minimal explanation.

・The authors compare the pathways enriched in GP1-3 in Qu et al and CCA1-2 in this paper, but did they check the overlap in protein levels? Are the same proteins up-regulated?

・In ccRCC, the authors found that proteins in metabolic pathways, translation, adhesion related to cytoskeleton, splicesome, nucleosome were enriched in CCA1, and those in focal adhesion, extracellular matrix, collagen organization were enriched in CCA2. How are these pathways important in ccRCC? How are they likely to contribute to the disease at individual level? I suggest to add more discussion about the link between the characteristics at a molecular level they found and that at individual level.

・I did not see any description on data availability. It is strongly recommended that the data be made public in some way.

Minor points

・I think it is not common to abbreviate "consensus clustering algorithm" as "CCA" and it is confusing with "canonical correlation analysis". How about abbreviate consensus clustering as CC?

・(In Fig1) It is recommended to avoid the combination of green and red (if possible) because it is not color blind-friendly. I could not read the figure under the color bar.

・On page 5, DFS and OS suddenly appear. Abbreviations used for the first time in a paper should be explained. For example, "disease free survival (DFS)" and "overall survival (OS)" should be used.

・The Mev software used for SAM is not cited.

([22,23] are references to TM4 and SAM)

Reviewer 3 Report

I would like to congratulate the authors for their work. However, I fell that the current paper could be improved before publication.

I believe it would be interesting to see the proteomics characterisation of metastatic RCC. As we know, in non metastatic RCC, current treatment of choice is surgery, and so, proteomics characterisation might not be of utmost interest for clinical practice. However, in the metastatic setting, novel treatments are emerging. I believe here could be a point of interest of your work.

Thank you and good luck!

Round 2

Reviewer 1 Report

I have no more comment.